# Effect of irrigation with treated wastewater on bermudagrass (*Cynodon dactylon* (L.) Pers.) production and soil characteristics and estimation of plant nutritional input

Mario Licata[1], Davide Farruggia[1], Nicolò Iacuzzi[1,2]*, Claudio Leto[1,2], Teresa Tuttolomondo[1], Giuseppe Di Miceli[1]

1 Department of Agricultural, Food and Forest Sciences, Università degli Studi di Palermo, Palermo, Italy,
2 Research Consortium for the Development of Innovative Agro-environmental Systems (CoRiSSIA), Palermo, Italy

* nicolo.iacuzzi@unipa.it

## Abstract

In recent years, climate change has greatly affected rainfall and air temperature levels leading to a reduction in water resources in Southern Europe. This fact has emphasized the need to focus on the use of non-conventional water resources for agricultural irrigation. The reuse of treated wastewater (TWW) can represent a sustainable solution, reducing the consumption of freshwater (FW) and the need for mineral fertilisers. The main aim of this study was to assess, in a three-year period, the effects of TWW irrigation compared to FW on the biomass production of bermudagrass [*Cynodon dactylon* (L.) Pers.] plants and soil characteristics and to estimate the nutritional input provided by TWW irrigation. TWW was obtained by a constructed wetland system (CWs) which was used to treat urban wastewater. The system had a total surface area of 100 m$^2$. An experimental field of bermudagrass was set up close to the system in a Sicilian location (Italy), using a split-plot design for a two-factor experiment with three replications. Results highlighted a high organic pollutant removal [five days biochemical oxygen demand (BOD$_5$): 61%, chemical oxygen demand (COD): 65%] and a good efficiency in nutrients [total nitrogen (TN): 50%, total phosphorus (TP): 42%] of the CWs. Plants irrigated with TWW showed higher dry aboveground dry-weight (1259.3 kg ha$^{-1}$) than those irrigated with FW (942.2 kg ha$^{-1}$), on average. TWW irrigation approximately allowed a saving of 50.0 kg TN ha$^{-1}$ year$^{-1}$, 24.0 kg TP ha$^{-1}$ year$^{-1}$ and 29.0 kg K ha$^{-1}$ year$^{-1}$ on average with respect to commonly used N-P-K fertilisation programme for bermudagrass in the Mediterranean region. Soil salinity increased significantly ($p \leq 0.01$) over the years and was detected to be higher in TWW-irrigated plots (+6.34%) in comparison with FW-irrigated plots. Our findings demonstrate that medium-term TWW irrigation increases the biomass production of bermudagrass turf and contributes to save significant amounts of nutrients, providing a series of agronomic and environmental benefits.

**Data Availability Statement:** All relevant data are within the manuscript and its Supporting Information files.

**Funding:** This study was funded by the Assessorato Regionale dell'Agricoltura, dello Sviluppo Rurale e della Pesca Mediterranea, Regione Sicilia (in English: Ministry of Agriculture, Rural Development and Mediterranean Fisheries of Sicilian Region) under project title "Contributo Enti di cui all'art. 128 della LR 11/2010 e SS.MM.ED I. - Anno 2021 – Fitodepurazione (Grant No. D73C21000060005). The funder did not play any role in the study design, data collection and analysis, decision to publish, or preparation of the manuscript.

**Competing interests:** The authors have declared that no competing interests exist.

# Introduction

In recent times, the increasing demand for water resources in arid and semi-arid countries in different production sectors has encouraged scientists and technicians to seek new strategies in order to ensure constant supply of water throughout the year, regardless of meteorology [1–3]. Agriculture is the production sector with the largest consumption of water in the world, accounting for 70% of total freshwater (FW) withdrawals on average [4]. Commonly used conventional water resources are estimated not to be sufficient to meet the water demand in these areas, as reported in various studies [1, 5–7]. Climate change has also contributed to creating long periods of water shortage in agricultural areas, emphasizing the need to better manage water resources. In this context, non-conventional water resources generated by specialized processes, such as salt water desalination or wastewater (WW), provide an ideal source of water for irrigation in agriculture [8–11]. For a series of reasons, treated wastewater (TWW), in particular, is an attractive form of sustainable water management for irrigation purposes. It increases water resources in the agricultural sector, providing lower quality water for irrigation and preserving high quality water for human consumption [12, 13]. It meets growing water demand and reduces the discharge of wastewater into soils and water bodies [11, 14]. It represents a source of mineral and organic nutrients and its application can increase crop yields and reduce the need for chemical fertilisers [2, 11, 14–17]. However, despite these benefits, public opinion tends to oppose water reuse projects, considering TWW as an unsafe solution due to the presence of pathogens in the water [18, 19]. Furthermore, some studies report that the medium- and long-term application of TWW irrigation could increase the level of salts in plants and soil, causing significant damage to the agro-ecosystems [2, 15, 20, 21]. As a consequence, efficient WW treatment is needed in order to safeguard the environment and, at the same time, provide benefits for agriculture. In some Italian regions, conventional treatment systems are outdated and do not perform all treatments needed to ensure high water quality. In this context, constructed wetlands (CWs) can play a key role in WW treatment as they can be integrated into conventional treatment plants (as tertiary-treatment technology) to complete the purification process of WW. A number of studies demonstrate the potential application of CWs in the agricultural sector to provide TWW for the irrigation of open field and horticultural crops, such as giant reed, eggplant and tomato [15, 16, 22–24]. Some studies focused on the effects of TWW on turfgrass species, evaluating how TWW affects plant and soil characteristics both in the short and long-term [2, 15, 25, 26]. It has been reported [27] that turfgrasses may be the best plants for reclaimed water irrigation due to greater absorption of nutrient contained in WW in comparison with that of other plants. Moreover, it has been observed [27] that most soil and plant-related problems concerning the use of WW may have a lower environmental impact on turfgrasses than on food crops. In the Mediterranean area, warm-season turf species are of great interest due to their high tolerance to drought and salt [28, 29], displaying low water needs and high recovery rates from other abiotic and biotic stresses [30–34]. Bermudagrass [*Cynodon dactylon* (L.) Pers.], for example, is the most widely-used warm-season species in the world for the creation of high quality turfs, such as those found in golf courses and athletics fields, and low quality ones, such as those found in public gardens and parks [34, 35]. In Italy, a number of studies have investigated the morphological, production and qualitative characteristics of native and improved bermudagrass accessions, highlighting high genetic variability and good adaptation to a wide range of soil and climate conditions [35–39].

However, despite the considerable interest in this species, little information is available on the use of TWW for the irrigation of bermudagrass. In Brazil, the authors [40] studied the effect of irrigation with unconventional water on productivity and quality of bermudagrass

and affirmed that the potential utilisation of secondary-treated sewage effluent could efficiently substitute potable water and saved a relevant part of the recommended nitrogen rate. Similar results were obtained by Nogueira et al. [41] who found high savings in nitrogen fertilisers and observed that the main benefits of TWW irrigation occurred in drought seasons. In a study carried out in Portugal [42], the authors found a clear and pronounced influence of the nutrient concentration in wastewater on the response of bermudagrass irrigated under several water regimes and saline conditions. Other studies [2, 29] confirm these findings but highlight that several turfgrass species could be negatively affected by TWW irrigation because of the sodium content. These studies highlight the benefits of TWW irrigation in terms of nutrients supply but do not investigate aspects relating to TWW production. Traditional grey infrastructures, mostly based on concrete, have been largely used to obtain TWW over time but their efficiency has been not always high mainly due to economic and technological reasons [43]. In this context, in many arid and semi-arid regions of the world, CWs contribute to the integrated water management, can efficiently purify various type of wastewater and provide TWW for irrigation purpose in accordance with sustainability criteria. As a consequence, irrigation with TWW obtained from CWs could become a regular practice in the Mediterranean area, also as regards turfgrass management, in order to save water and reduce fertiliser uses. Considering the high pollutants removal efficiency of a CW and taking into consideration the fact that TWW represent a source of water and nutrients, the initial hypothesis of this study was that TWW obtained by CW can greatly integrate the conventional irrigation and fertilisation programmes for the maintenance of bermudagrass turf, providing agronomic and environmental benefits. With this in mind, the aims of this paper were: 1) to assess the effects of irrigation with TWW obtained from a pilot scale horizontal sub-surface flow system (HSSFs) CW on the morphological, production and qualitative characteristics of bermudagrass turf and on the chemical soil properties in the medium-term, 2) to estimate nutritional input provided by TWW irrigation.

## Materials and methods

### Test site

Tests were carried out in the three years from 2016 to 2018 in the experimental area of the pilot HSSFs CW in Raffadali (Italy), a rural municipality in the South-West of Sicily (37˚59'56"40 N–13˚16'50"16 E, 740 m a.s.l.). The HSSFs CW was used to treat urban WW produced by a municipal treatment plant. An experimental field of bermudagrass plants was set up close to the pilot HSSFs CW. According to the Köppen–Geiger climate classification [44], the study area is characterized by a warm temperate climate with dry summers. The annual average rainfall is approximately 650 mm, mainly distributed between October and April. With reference to time series 1982–2012, the annual average temperature was 17.5˚C, the average maximum temperature was 23.5˚C and the average minimum temperature was 11.2˚C.

### Description of the test HSSFs CW

The HSSFs CW was located in an urban park (S1 Fig). It included two separate, parallel units each 50 m long and 1 m wide, with a total surface area of 100 $m^2$. The units were built in concrete and lined with sheets of ethylene and vinyl-acetate. Filter bed depth was 0.50 m with a water depth of 0.30 m and a 2% slope. The substrate was made of evenly sized 30 mm silica quartz river gravel (Si 30.0%; Al 5.11%; Fe 6.10%; Ca 2.65%; Mg 1.05%) with a porosity of 35–40%. In February 2008, the two units were separately planted with giant reed (*Arundo donax* L.) at a density of 4 rhizomes $m^{-2}$ and umbrella sedge (*Cyperus alternifolius* L.) at a density of 5 stems $m^{-2}$. Two separated monoculture crop systems were, then, established in the pilot plant.

The HSSFs CW was fed with pre-treated urban WW from the sewage treatment system which carried out primary and secondary treatments. WW was fed initially into a 15 m$^3$ water-proofed, vibrated cement storage tank. The tank was equipped with a litre gauge and an outlet valve for the periodic cleaning of solid sediments. WW was fed into a static degreaser to separate fats and organic wastes and pumped through a perforated polyvinylchloride pipe into the two CW units. The pipe was placed 10 cm from the surface of the substrate. WW was homogeneously distributed in each unit through a timer-controlled pumping system. Pumping was continuous throughout the day without variations in time. TWW was, then, collected using a perforated drainage pipe system, placed at the bottom of the filter bed and conducted downhill into a system of four interconnected tanks of 5 m$^3$ each (S2 Fig). The last of these tanks was used to supply water for irrigation purposes and connected to a sprinkler irrigation system. The two units were tested using a hydraulic loading rate (HLR) of 6 cm day$^{-1}$ and hydraulic retention time (HRT) of 8.30 days.

## Urban wastewater analysis

WW samples (S) were taken monthly at the inlet and outlet pipes from April to September of each year. Sampling amounted to a total of 72 events (36 per planted-unit). 1 litre (L) of WW was collected at each sampling point. The influent sample was taken close to the pipe while the effluent sample was collected at the mouth of the outflow pipe. The influent and effluent samples were instantaneous samples. Samples were collected using high polyethylene density (HDPE) bottles. The pH value, electrical conductivity of water (EC$_w$), temperature (T) and dissolved oxygen (DO) levels were determined directly on site using a portable Universal meter (Multiline WTW P4). Total suspended solids (TSS), five days biochemical oxygen demand (BOD$_5$), chemical oxygen demand (COD), total nitrogen (TN), ammonia nitrogen (NH$_4$-N), total phosphorus (TP), sodium (Na$^+$), potassium (K$^+$), calcium (Ca$^{++}$) and magnesium (Mg$^{++}$) levels were determined according to Italian water analytical methods [45]. Total coliforms (TC), faecal coliforms (FC), faecal streptococci (FS), *Escherichia coli* (E. coli) and *Salmonella* spp. levels were determined according to standard methods for water testing [46]. For each planted-unit, pollutant RE was based on pollutant concentration and calculated in accordance with International Water Association [47]:

$$RE = \frac{(C_i - C_0)}{C_i \times 100} \tag{1}$$

where C$_i$ and C$_0$ are the mean concentrations of the pollutants in the influent and effluent. The TWW was, then, used to irrigate bermudagrass plants.

## HSSFs CW water balance

Reference evapotranspiration (ET$_0$) was calculated using the FAO Penman-Monteith method [48]. In each planted unit, the water balance was estimated using the equation provided by International Water Association [36]:

$$Q_0 = Q_i(P - ET_c)A \tag{2}$$

where Q$_o$ = output wastewater flow rate (m$^3$ day$^{-1}$), Q$_i$ = wastewater inflow rate (m$^3$ day$^{-1}$), P = precipitation rate (mm day$^{-1}$), ET$_c$ = crop evapotranspiration (mm day$^{-1}$), A = wetland top surface area (m$^2$). The amount of water at the inlet and outlet of each unit was determined with a volumetric flow meter. Rainfall was determined with a pluviometer. In this study, Q$_i$ was constant for all study periods (60 m$^3$ 10-day$^{-1}$) depending on the technical characteristics of the HSSFs CW. For each unit, the water balance was calculated separately every 10 days

during the period May to September, in correspondence with the growth dynamics of the two macrophytes.

## Bermudagrass experimental field

A seeded bermudagrass variety, Princess 77, was used for the tests. The date of sowing was 11 May 2015. The plots were 4 m$^2$ and were spaced 50 cm apart. The inter-plot spaces were periodically treated with glyphosate [N-(phosphonomethyl) glycine] at 4 kg ha$^{-1}$ year$^{-1}$ in order to avoid the spread of plants between plots. One year before the date of sowing, the experimental area was treated with the same herbicide twice a year at 2.88 kg ha$^{-1}$ in order to minimize weed competition. The soil was clay loam (40% sand, 21% silt and 39% clay) and was classified as brown soil and regosols (World Reference Base for Soil Resources). In particular, the soil had a pH value of 7.60, a cation exchange capacity of 33.8 meq 100 g$^{-1}$, a total CaCO$_3$ of 1.30 g kg$^{-1}$, a total N of 1.20 g kg$^{-1}$ and a K content of 530 ppm ± 1, on average. Before sowing bermudagrass, the soil nutrients were evaluated.

A split-plot design for a two-factor experiment was adopted with three replications. The main plot factor was year (Y) with three treatment levels: $Y_1$ (2016), $Y_2$ (2017), $Y_3$ (2018). The sub-plot factor was irrigation water (IW) with three treatment levels: $IW_1$ (FW); $IW_2$ (TWW from giant reed-planted unit); $IW_3$ (TWW from umbrella sedge-planted unit).

In 2015, the experimental field was equipped with a sprinkling irrigation system and was irrigated with FW in order to establish the turfgrass. In 2016–2018, irrigation was applied from May to September three times per week, on average, both with FW and TWW in order to maintain active growth of the turfgrass. Irrigation events were effectuated supplying 5000 m$^3$ ha$^{-1}$ year$^{-1}$ of water (80 m$^3$ ha$^{-1}$ of water during each event) and were scheduled during intense plant growth in the spring and summer months. Bermudagrass water needs were determined by calculating the difference between the amount of water lost due to evapotranspiration and rainfall rates. A weather station, belonging to the Sicilian Agrometeorological Information Service [49], was used to collect daily minimum and maximum air temperatures and total rainfall. Irrigation volume was calculated in accordance with the following equation:

$$V = 10,000 \times (FC - WP) \times \phi \times H \tag{3}$$

where 10,000 is the area of 1 hectare, FC is the soil water content at field capacity, WP is the soil water content at wilting point, $\phi$ is the bulk density of soil and H is the height of the soil layer from wet, equivalent to rooting depth of bermudagrass.

In 2015, plots received 50.0 kg N ha$^{-1}$, 20.0 kg P$_2$O$_5$ ha$^{-1}$ and 40.0 kg K$_2$O ha$^{-1}$ per month of growth from May to September. In 2016–2018, the FW-irrigated plots were managed with the same N, P and K fertilisation programme used in the previous year. In the TWW-irrigated plots, we estimated the amounts of N, P and K supplied by irrigating with TWW in order to integrate the nutrient need of bermudagrass, based on previous analyses. Both for FW and TWW, the Sodium Adsorption Ratio (SAR) was calculated considering the square root of Na$^+$ to Ca$^{++}$ plus Mg$^{++}$, as described by Lesch and Suarez [50].

Concerning other practices, the turf was maintained at a mowing height ranging from 30 to 35 mm and was mowed using a helicoidal mower during the vegetative stage. Mowing was carried out twice per week during intense growth periods with the subsequent removal of grass clippings. In 2016–2018, weeds were manually removed. No insecticide and fungicide treatments were carried out during the test period.

**Plant measurements.** For each treatment, the main morphological, production and qualitative parameters of bermudagrass were considered. Leaf width was determined monthly from April to October by randomly removing 100 flattened leaves per subplot and measuring the

leaf width at a distance of 1 cm from its ligule [36]. Shoot density was calculated in June and September by counting the number of shoots in 50 $cm^2$ core that was collected to a depth of 30 cm and close to the plot centre, where the turf was assumed to be fully established [36]. Turf colour determination was based on a 1 (= light green) to 9 (= dark green) visual rating scale after mowing [37]. Visual turf quality was based on a 1 (= poorest or dead) to 9 (= outstanding or ideal) visual rating scale [35]. Turf quality was based on colour, leaf texture, uniformity of coverage and shoot density. Turf colour and quality were determined monthly during the vegetative growth of bermudagrass. Above-ground dry biomass was calculated by removing all plant tissue from the core top and drying the collected material in an oven at 60° to constant weight [36]. A grass sample was taken randomly in each subplot of each treatment level of irrigation. Sampling was carried out in June and September.

**Soil sampling and analysis.**   Before planting, three sampling spots per plot were randomly combined for the plot sample and analysed. Three soil samples were taken at a depth of 0–20 cm from each plot, close to the rhizosphere of bermudagrass. Undisturbed soil samples were collected using hand augers from a vertical boring, mixed, placed in clean polyethylene bags and labeled. The same procedure was carried out after each irrigation period. Soil samples were air dried, ground and sieved to pass through a 2-mm sieve screen and analyzed for chemical and physical characteristics.

The samples were analyzed for pH and EC in the ratio of 1:2 dry soil: water extract, pH was determined with a calibrated pH-meter (± 0.01), EC with a calibrated conductivimeter (% 0.05 of value), total organic carbon (TOC) of soil with the Walkley and Black method [51] (± 0.01, %), total nitrogen (TKN) by the Kjeldahl procedure [52] (± 0.02, g $kg^{-1}$), assimilable P by the Olsen method [53] (± 0.02, ppm) and total calcium carbonate using the Drouineau method [54] (± 0.01, %). $K^+$ (± 0.08, ppm), $Mg^{++}$ (± 0.09, ppm) and $Na^+$ (± 0.09, ppm) contents were determined by atomic absorption spectrophotometer. All the analyses were carried out at the Corissia Research Center lab in Palermo (Italy).

**Weather data.**   The weather station was situated close to the pilot HSSFs CW. The station was equipped with a MTX datalogger (model WST1800) and various weather sensors. Various sensors provided data on daily minimum and maximum air temperatures and rainfall and allowed us to calculate $ET_0$.

## Statistical analyses

Statistical analysis was performed using the package MINITAB 19 for Windows. For TWW composition, all the representative values were shown using mean ± standard deviation calculations. For plant and soil parameters, analysis of variance (F-test; p < 0.01) was carried out and the difference between means was investigated out using the Tukey test (p < 0.01).

## Results and discussions

### Rainfall and air temperature trends in the experimental area

Trends of average maximum and minimum air temperatures and total rainfall during 2016–2018 are shown in Fig 1. The average maximum air temperature was 23.8°C, while the average minimum air temperature was 11.1°C. In each year, maximum and minimum air temperatures increased from March to August and decreased up to the end of December. The highest maximum air temperatures were recorded in July and August while the lowest were recorded in January and February. Total rainfall ranged from 494.6 mm (2016) to 904.9 mm (2018). Average rainfall during the 3-year period was 637.7 mm. The highest total rainfall levels occurred during autumn and winter. In summer, the highest total rainfall (101.6 mm) was

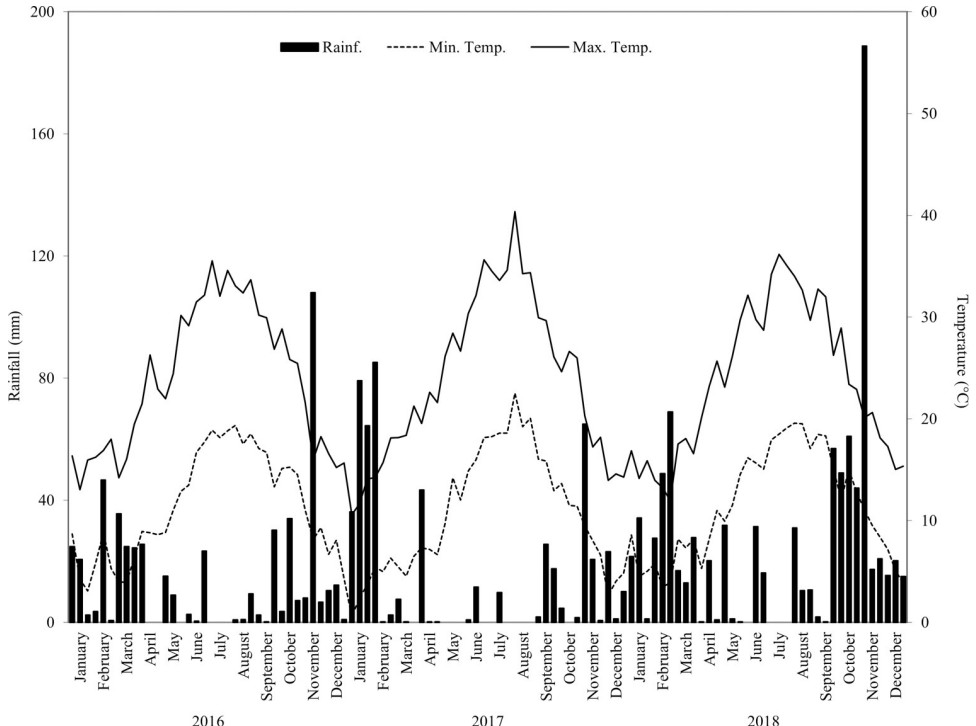

**Fig 1. Rainfall and air temperature trends during the test period in the experimental area.** For each month, three 10-day periods are shown.

recorded in 2018 while the lowest (37.4 mm) in 2016. The average 3-year rainfall level in summer was 62.2 mm (S1 Table).

In particular, observing the growth season of bermudagrass plants (from April to September), average air temperatures and rainfall levels differed slightly from one year to the next. Irrigation volumes differed over the 3 years due to different evapotranspiration rates.

## HSSFs CW performance

Data showing the chemical-physical variations relating to WW are shown in Table 1. At the inlet of the HSSFs CW, the average pH value was moderately alkaline and significantly different with respect to that recorded at the outlet. In particular, average effluent pH values were

**Table 1. Variation (VA) of pH, T and ECw in the two HSSFs CW units from April to September during 2016/2018.** For each unit, three-year average values (± standard deviation) are shown (n = 36).

| Parameter | Influent | Effluent[1] | Effluent[2] | VA (%)[1] | VA (%)[2] | Threshold values for reuse of TWW[3] | t-Test[4] |
|---|---|---|---|---|---|---|---|
| pH | 15.1 ± 0.12 | 14.8 ± 0.1 | 14.7 ± 0.07 | 1.91 | 2.77 | - | * |
| T (˚C) | 7.41 ± 0.50 | 7.10 ± 0.10 | 7.0 ± 0.42 | 5.11 | 5.45 | 6–9.5 | * |
| $EC_w$ (µS cm$^{-1}$) | 0.51 ± 22.1 | 0.58 ± 21.1 | 0.56 ± 22.1 | 16.2 | 12.9 | 3 | * |
| DO (mg L$^{-1}$) | 1.21 ± 0.02 | 1.01 ± 0.02 | 0.99 ± 0.01 | 16.5 | 18.2 | - | |

[1] Effluent from giant reed-unit.

[2] Effluent from umbrella sedge-unit.

[3] Threshold values for Italian Decree 152/2006 concerning irrigation purpose.

[4] Significant (*) differences between influent and effluent values (p < 0.01).

**Table 2. Main chemical composition of the treated wastewater from inlet to outlet of the HSSFs CW.** Removal efficiency (RE) from April to September 2016/2018. For each unit, three-year average values (± standard deviation) are shown (n = 36).

| Parameter | Influent | Effluent[1] | Effluent[2] | RE (%)[1] | RE (%)[2] | Threshold values for reuse of TWW[3] | t-Test[4] |
|---|---|---|---|---|---|---|---|
| TSS (mg L$^{-1}$) | 39.8 ± 11.4 | 12.4 ± 9.11 | 13.7 ± 8.79 | 69.7 | 65.6 | 10 | * |
| BOD$_5$ (mg L$^{-1}$) | 32.5 ± 4.7 | 12.5 ± 2.82 | 12.7 ± 2.85 | 61.5 | 60.9 | 20 | * |
| COD (mg L$^{-1}$) | 55.6 ± 8.5 | 18.6 ± 2.55 | 19.1 ± 1.89 | 66.5 | 65.6 | 100 | * |
| TN (mg L$^{-1}$) | 21.3 ± 2.5 | 10.4 ± 1.63 | 10.5 ± 1.55 | 51.2 | 50.6 | 15 | * |
| TP (mg L$^{-1}$) | 8.62 ± 0.57 | 4.90 ± 0.43 | 5.21 ± 0.66 | 43.0 | 39.1 | 2 | * |

[1] Effluent from giant reed-unit.

[2] Effluent from umbrella sedge-unit.

[3] Threshold values for Italian Decree 152/2006 concerning irrigation purpose.

[4] Significant (*) differences between influent and effluent values (p < 0.01).

found to be less alkaline than influent values and ranged from 6.31 to 7.72. These results were in accordance with the findings of Kadlec and Knight [55] and can be explained by the production of $CO_2$ caused by the decomposition of plant residues in the substrate in the medium term, the removal of various components of WW retained in the root area and the nitrification of ammonia [56]. In the case of EC, the effluent values were significantly higher than influent ones probably due to ET processes which caused substantial water loss in the CW, increasing the solute concentration in the solution. ET rates were different in the two units depending on the morphological and physiological characteristics of the two macrophytes. Previous studies carried out in this field confirmed our findings [57–60]. DO levels at the outflow were approximately equal to 1.0 mg L$^{-1}$, consistent with values found in other HSSFs [47]. No significant differences were found in the two planted units despite the diverse root apparatus of macrophytes.

When observing the main chemical composition of TWW between influent and effluent (S2, S3 Tables), all parameters in the study showed significant differences in the two planted units (Table 2, Fig 2). Average TSS levels were different in the two planted units highlighting the effect of species on removal processes [15]. In the giant reed-unit, the higher plant and root density affected TSS filter mechanisms, leading to lower levels of sedimentation at the roots and substrate. In both planted units, TSS effluent levels were not always within the threshold values imposed by Italian Decree No. 152/2006 regarding the reuse of TWW for irrigation purposes. This depended on various factors, such as plant growth trends, WW composition and type of WW pre-treatment over the seasons. Observing the TSS RE values, these were consistent with those found on literature for HSSFs CW. In both effluents, BOD$_5$ and COD average levels were found below 20 mg L$^{-1}$ in accordance with the legal limits of the Italian Decree. The high removal rates of BOD$_5$ and COD were due to a combined action of microorganisms, plants and substrate, as well-documented in literature [47]. BOD$_5$ RE varied from 60.9 (umbrella sedge-unit) to 61.5% (giant-reed unit) while COD RE ranged from 65.6 (umbrella sedge-unit) to 66.5% (giant-reed unit). In general, the BOD$_5$ and COD RE values stayed within a range consistent with previous HSSFs CW studies treating urban WW [61]. Significant differences were found between influent and effluent concerning TN and TP, the removal efficiencies of which were found to be similar in both planted-units. TN RE was recorded as lower in comparison with organic compound removal due to lower oxygen levels in the system, which greatly affected the ammonia nitrification. This represents, in fact, one of the most important nitrogen removal mechanisms in a HSSFs CW, as sustained by a number of authors. For TP, RE values were much lower than those of TSS, BOD$_5$, COD and TN due to a range of

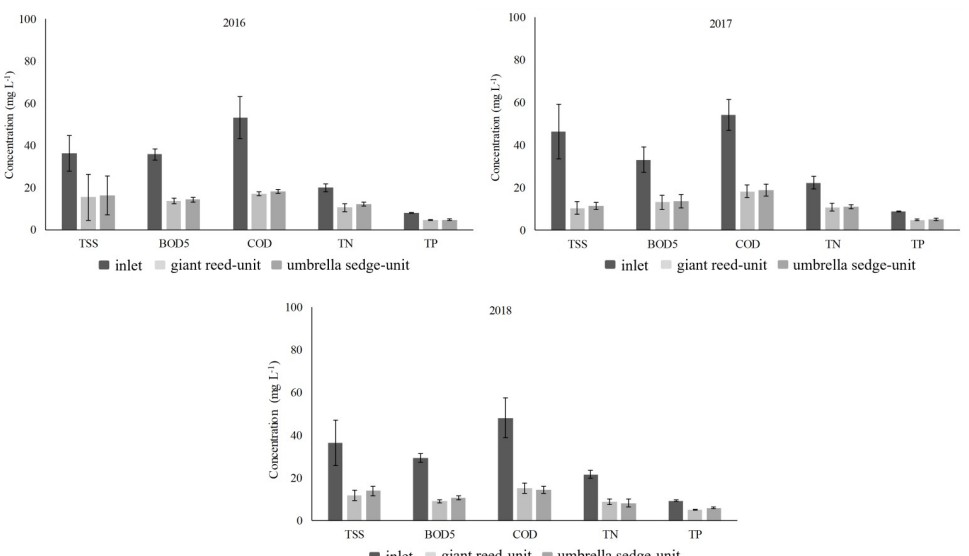

**Fig 2. Average concentrations for chemical parameters at the HSSFs CW inlet and outlet in 2016, 2017 and 2018.** Bars represent standard deviation.

factors, such as the presence of undecomposed plant parts around the substrate surface and the granular saturation of most of the substrate sorption sites, as reported by Maelhum et al. [62]. In both effluents, TP levels exceeded on average the legal limits established by the Italian Decree.

On a microbiological level (Table 3, Fig 3), TC, FC, FS and E. coli average levels showed significant differences between influent and effluent (S2, S3 Tables). In the giant reed- and umbrella sedge-units, average RE levels were recorded to be above 80% in accordance with the findings of previous studies carried out under similar operating conditions at the HSSFs CW. In 2016–2018, data obtained for *Escherichia coli* at outlet of the two planted-units were not always found to be within the limits imposed by Italian Decree No. 152/2006, this result mainly depended on initial microbial concentrations in WW and the phenological stages of plants.

**Table 3. Main microbiological composition of the treated wastewater from inlet to outlet of the HSSFs CW.** Removal efficiency (RE) from April to September 2016/2018. For each unit, three-year average values (± standard deviation) are shown (n = 36).

| Parameter | Influent | Effluent[1] | Effluent[2] | RE (%)[1] | RE (%)[2] | Threshold values for reuse of TWW[3] | t-Test[4] |
|---|---|---|---|---|---|---|---|
| TC (CFUs 100 ml$^{-1}$) | 4.44 ± 0.06[5] | 3.44 ± 0.10 | 3.48 ± 0.04 | 89.7 | 88.8 | - | * |
| FC (CFUs 100 ml$^{-1}$) | 4.28 ± 0.06 | 3.33 ± 0.05 | 3.37 ± 0.06 | 88.9 | 88.2 | - | * |
| FS (CFUs 100 ml$^{-1}$) | 3.98 ± 0.04 | 3.24 ± 0.06 | 3.26 ± 0.06 | 81.8 | 80.2 | - | * |
| *Escherichia coli* (CFUs 100 ml$^{-1}$) | 3.15 ± 0.06 | 2.07 ± 0.08 | 2.09 ± 0.04 | 89.1 | 87.7 | 10 and 100[6] | * |
| *Salmonella* spp. (CFUs 100 ml$^{-1}$) | Absent | Absent | Absent | | | - | |

[1] Effluent from giant reed-unit.

[2] Effluent from umbrella sedge-unit.

[3] Threshold values for Italian Decree 152/2006 concerning irrigation purpose.

[4] Significant (*) differences between influent and effluent values (p < 0.01).

[5] The average concentration values are shown as units of $Log_{10}$.

[6] 10 CFUs 100 ml$^{-1}$ (80% of samples) and 100 CFUs 100 ml$^{-1}$ as maximum value point.

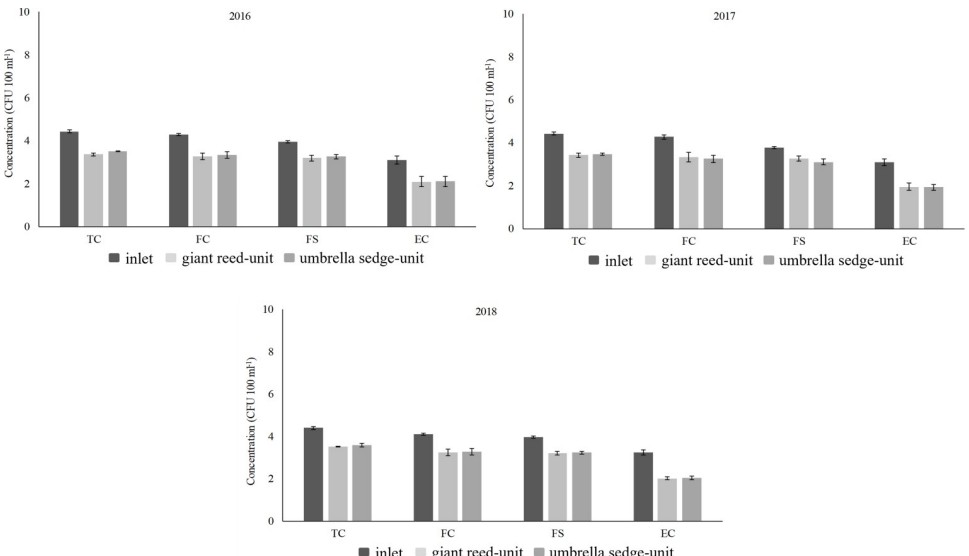

**Fig 3. Average concentrations for microbiological parameters at the HSSFs CW inlet and outlet in 2016, 2017 and 2018.** Bars represent standard deviation. The values are shown as units of $Log_{10}$.

## HSSFs CW water loss

Table 1 reports the WW inflow and outflow rates that were calculated in the two planted units of the HSSFs CW (Table 4). In general, ET processes affected the amount of TWW at the outflow of the CWs. As $Q_i$ was constant for all of the 10-day periods in the study (based on the technical and hydraulic characteristics of the HSSFs CW), $Q_o$ differed as result of a different intensity of ET detected in the planted units (S4 Table).

ET was influenced by the morphological and physiological characteristics of the species and climate conditions, in particular. In fact, some factors, such as leaf size, number of stomata per $cm^2$ of leaf and number of plants per $m^2$ of the species and air temperature, solar radiation and moisture levels, greatly affected the water loss rates in the planted units over the seasons. Considerable water loss was observed during summer due to higher ET processes for the same period. However, despite this, it is important to note that a large amount of TWW was obtained at the outlet of the planted units in the same period. Therefore, we can sustain that HSSFs CW allows us to obtain TWW leading to savings in FW during summer months. These results confirm the findings of previous studies in this field and highlight that a CW can

**Table 4. Wastewater inflow and outflow rates in the HSSFs CW during summer.** Average values of 3-year tests are shown.

| Month | $Q_i$ (m$^3$ 10-days$^{-1}$) | $Q_o$ (m$^3$ 10-days$^{-1}$)[1] | $Q_o$ (m$^3$ 10-days$^{-1}$)[2] |
|---|---|---|---|
| May | 60.0 | 55.8 | 56.5 |
| June | 60.0 | 50.0 | 51.3 |
| July | 60.0 | 41.8 | 43.8 |
| August | 60.0 | 43.9 | 46.1 |
| September | 60.0 | 43.9 | 50.2 |
| *Total* | 300.0 | 235.5 | 247.9 |

[1] $Q_o$ from giant reed-unit.

[2] $Q_o$ from umbrella sedge-unit.

guarantee continuous water availability for irrigation purposes in regions with prolonged water scarcity [22, 25].

## Freshwater and treated wastewater characteristics

Table 5 shows the main chemical characteristics of FW and TWW used for irrigation of bermudagrass. In general, TWW had higher average values of mineral and organic compounds than FW. When considering the microbiological levels in irrigation water, *Escherichia coli* was only detected in TWW. Comparing the chemical composition of FW and TWW from May to September, a significant change over the study period was found. The lowest variations of nutrients between the two sources of irrigation water were observed during summer due to higher pollutant RE of the HSSFs CW. For example, observing the TKN level of the effluents from June to September, a considerable decrease in N in WW was recorded mainly due to higher N plant uptake and microbial processes. On the contrary, during autumn, N levels increased more in WW than FW because of a lower nitrification rate and plant uptake intensity. As stated in literature, the qualitative characteristics of TWW are fundamental for turfgrass growth. N, P and K play an important role in metabolic processes and morphological development of plants, thus their concentration values in TWW should be strongly considered because of the effects on plant and soil. Furthermore, Ca, Mg and Na are also important due to effects on physiological processes within the plants, as noted in previous studies [2, 26].

Many authors agree on the importance of TWW as a source of irrigation water for bermudagrass but highlight that it depends on the type and amounts of dissolved salts and nutrients and how much TWW is used [27, 32]. In the medium- and long-term, it has been demonstrated that the prolonged use of TWW with high levels of organic and mineral compounds can determine negative effects on bermudagrass growth and soil properties. For example, it has been reported [63] that a number of physiological processes in plants such as

**Table 5. Main chemical and microbiological characteristics of freshwater (FW) and treated wastewater (TWW) that were used for irrigation.** Average values (± standard deviation) are shown (n = 20).

| Parameter | FW | TWW[1] | TWW[2] | Threshold Values for Italian Decree No. 152/2006 |
|---|---|---|---|---|
| pH | 7.00 ± 0.01 | 7.10 ± 0.10 | 7.01 ± 0.42 | 6–9.5 |
| EC ($\mu$S cm$^{-1}$) | 253.2 ± 1.56 | 580.1 ± 21.1 | 560.3 ± 22.1 | 3000 |
| DO (mg L$^{-1}$) | Not available | 1.01 ± 0.01 | 0.99 ± 0.02 | - |
| TSS (mg L$^{-1}$) | Not detected | 12.4 ± 1.90 | 13.7 ± 1.83 | 10 |
| BOD$_5$ (mg L$^{-1}$) | 1.21 ± 0.03 | 12.1 ± 0.59 | 6.5 ± 0.53 | 20 |
| COD (mg L$^{-1}$) | 1.87 ± 0.07 | 18.6 ± 0.55 | 19.1 ± 1.89 | 100 |
| TN (mg L$^{-1}$) | Not available | 10.4 ± 1.63 | 10.5 ± 0.55 | 15 |
| NO$_3$N (mg L$^{-1}$) | 0.26 ± 0.02 | 2.52 ± 1.55 | 2.88 ± 1.21 | - |
| TP (mg L$^{-1}$) | 0.47 ± 0.04 | 0 ± 0.36 | 5.21 ± 0.66 | 2 |
| K (mg L$^{-1}$) | 2.07 ± 1.01 | 56.4 ± 3.52 | 62.2 ± 4.9 | - |
| Ca (mg L$^{-1}$) | 19.5 ± 1.12 | 72.3 ± 2.10 | 68.1 ± 2.26 | - |
| Na (mg L$^{-1}$) | 11.2 ± 0.68 | 145.1 ± 2.11 | 156.3 ± 1.18 | - |
| Mg (mg L$^{-1}$) | 14.6 ± 1.22 | 23.0 ± 1.01 | 19.2 ± 0.87 | - |
| SAR (meq L$^{-1}$) | 2.71 ± 1.02 | 3.81 ± 0.51 | 4.32 ± 0.31 | - |
| *Escherichia coli* (CFU 100 ml$^{-1}$) | Not detected | 2.07 ± 1.08 | 2.09 ± 0.99 | 10 (80% of samples) and 100 (maximum value point) |

[1] TWW from giant reed-unit.

[2] TWW from umbrella sedge-unit.

photosynthesis, carbohydrate storage and rooting could be limited due to stress conditions caused by ion imbalances, ion toxicity and water deficiency resulting from high amounts of dissolved salts in WW.

In this study, the quality of irrigation water was assessed using the guidelines based on the findings of Westcot and Ayers and improved by McCarty [30], with reference to turfgrass (Table 6).

Furthermore, a series of restrictions on FW and TWW use in irrigation were observed in accordance with Castro et al. [2], as reported in Table 7. According to the guidelines, both FW and TWW were, in general, adequate for irrigation. Regarding TWW, pH values and average N levels were within recommended limits whilst average Na levels in the effluents of the HSSFs CW showed a degree of moderate restriction on use for irrigation. Average EC values for TWW from the giant reed- and umbrella sedge-units were not critical for plant growth. In the TWW-planted units, average values for parameters regarding salinity (EC) and infiltration (SAR) required no degree of limitation on use for irrigation. These results were consistent with those obtained by other authors for TWW after CWs treatment [1, 2, 15] and highlight the great suitability of these systems to improve the quality of non-conventional water.

## Morphological and qualitative traits and biomass production of bermudagrass plants

Year and irrigation water (S5 Table) produced significant differences for leaf width, shoot density and dry above-ground weight (Table 8). Results of analysis of variance indicated that the year-by-irrigation water interaction was not significant for all morphological and qualitative characteristics of bermudagrass in the study.

**Table 6. General guidelines for interpretation of water quality for turfgrass irrigation (Ayers and Westcot, 1985, modified from McCarty, 2001).**

| Item | | Minor problems | Increasing problems | Severe problems |
|---|---|---|---|---|
| *Soil permeability/infiltration* | | | | |
| EC (water) | (dS m$^{-1}$) | < 0.75 | 0.75–3.0 | > 3.0 |
| EC (soil) | (dS m$^{-1}$) | 2.0–4.0 | 4.0–12.0 | > 12.0 |
| Sodium (SAR) | (meq L$^{-1}$) | < 6.0 | 6.0–9.0 | > 9.0 |
| Total dissolved salts | (ppm) | < 450.0 | 450–2000 | > 2000 |
| Bicarbonates (HCO$_3$) | (ppm) | 0–120 | 120–180 | 180–600 |
| Resisual sodium carbonates | (meq L$^{-1}$) | ≤ 1.25 | 1.25–2.50 | > 2.50 |
| *Turf toxicity from root absorption* | | | | |
| Sodium | (meq L$^{-1}$) | < 3.0 | 3.0–9.0 | > 9.0 |
| Chloride | (meq L$^{-1}$) | < 2.0 | 2.0–10.0 | > 10.0 |
| Boron | (mg L$^{-1}$) | < 1.0 | 1.0–2.0 | > 2.0 |
| *Turf toxicity from foliar contact* | | | | |
| Sodium | (meq L$^{-1}$) | < 3.0 | > 3.0–9.0 | > 9.0 |
| Chloride | (meq L$^{-1}$) | < 3.0 | 3.0–10.0 | > 10.0 |
| Boron | (meq L$^{-1}$) | < 0.75 | 0.75–3.0 | > 3.0 |
| *Ornamental plant tolerance* | | | | |
| Ammonium-N (NH$_4$-N) | (mg L$^{-1}$) | < 5.0 | 5.0–30.0 | > 30.0 |
| Nitrate-N (NO$_3$-N) | (mg L$^{-1}$) | < 5.0 | 5.0–30.0 | > 30.0 |
| Bicarbonates (HCO$_3$) | (meq L$^{-1}$) | < 1.50 | 1.50–8.50 | > 8.50 |
| Unsightly foliar deposits | (mg L$^{-1}$) | < 90 | 90.0–520.0 | > 520.0 |
| Residual chlorine | (mg L$^{-1}$) | < 1.0 | 1.0–5.0 | > 5.0 |
| pH | normal range 6.0–8.40 | | | |

**Table 7. Restrictions on use of freshwater and treated wastewater obtained from HSSFs CW in irrigation.**

| Item | FW | TWW[1] | TWW[2] |
|---|---|---|---|
| Salinity (EC) | None | None | None |
| Infiltration (SAR) | None | None | None |
| *Specific ion toxicity*: | | | |
| Na | None | Moderate | Moderate |
| *Miscellaneous effects*: | | | |
| $NO_3$-N | None | None | None |

[1] TWW from giant reed-unit.

[2] TWW from umbrella sedge-unit.

Observing the quality parameters, visual quality and colour of turf did not significantly vary over the years and were not affected by the source of irrigation water, despite a high N content in the TWW. This was unexpected due to fact that N is an important component of chlorophyll molecules and affects plant growth, green leaf colour and, consequently, the visual appearance of turf [32]. The lack of significant differences between the irrigation treatments can be explained by the fact that, despite having differentiated fertilisation management programs, FW- and TWW-irrigated plots, received an appropriate amount of N throughout the test period, which contributed to maintaining the turfgrass in good growing condition.

Concerning herbaceous biomass production, it was greatly influenced by the source of irrigation water and increased over the years (Fig 4). Above-ground dry weight was greatly affected by climate conditions in the three-year period. In 2017, dry weight reached the maximum value (1207.7 kg ha$^{-1}$) (Table 4) due to differing average air temperatures, ET rate and lower rainfall during summer months in comparison with those of 2016 and 2018; this caused led to greater levels of irrigation water in that year.

In all harvests, dry above-ground biomass collected from TWW-irrigated plots was on average higher than that collected in FW-irrigated plots. This was confirmed by a number of authors [1, 64–66] who investigated the effect of irrigation with reclaimed water on grass characteristics. In a two-year study carried out in Spain, evaluating a grass mixture of tall fescue (*Festuca arundinacea* Schreb.) and Kentucky bluegrass (*Poa pratensis* L.), the authors [2] found that the growth of the WW-irrigated crops was always significantly higher than the control crop. In general, the increase in biomass production can be linked to the higher nutrient

**Table 8. Morphological, qualitative and production characteristics of bermudagrass plants in response to year and irrigation water during the study period.** Average values of 3-year tests are shown.

| Parameter | Leaf width (mm) | Shoot density (n cm$^{-2}$) | Visual quality (1–9) | Color (1–9) | Above-ground dry weight (kg ha$^{-1}$) |
|---|---|---|---|---|---|
| Year (Y) | | | | | |
| $Y_1$ | 1.48 B | 1.94 A | 6.51 A | 6.41 A | 1096.4 C |
| $Y_2$ | 1.49 B | 1.97 B | 6.55 A | 6.30 A | 1156.7 B |
| $Y_3$ | 1.52 A | 1.96 B | 6.56 A | 6.33 A | 1207.7 A |
| Irrigation Water (IW) | | | | | |
| $IW_1$ | 1.44 B | 1.88 B | 6.50 A | 6.30 A | 942.2 B |
| $IW_2$ | 1.52 A | 2.01 A | 6.52 A | 6.31 A | 1248.3 A |
| $IW_3$ | 1.54 A | 1.99 A | 6.53 A | 6.29 A | 1270.4 A |
| Y × IW | n.s. | n.s. | n.s. | n.s. | n.s. |

Means followed by the same letter in the same column are not significantly different according to Tukey's test (p ≤ 0.01). n.s. = not significant.

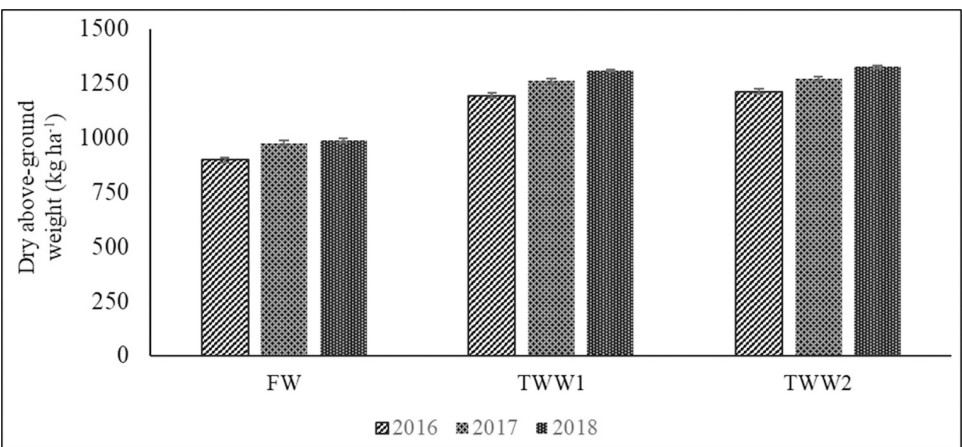

**Fig 4. Total herbaceous biomass production in freshwater- and treated wastewater-irrigated plots during 2016–2018.** Bars represent standard deviation.

levels amount received through TWW-irrigation in comparison with FW-irrigation [25]. Although this thesis is shared by a number of authors, it was observed [63] for some species that potable water obtained better results in terms of biomass production than reclaimed water. This mainly depends on the chemical composition of water; for example, a high $Na^+$ or heavy metal content in water could negatively affect plant biomass production due to direct effects on plant and soil characteristics. However, we also need to consider the ability of the species to tolerate high salts and nutrient content in water. For example, in the same study [63], the authors found significant differences between bermudagrass and bentgrass (*Agrostis stolonifera* L.), which produced different biomass levels under reclaimed water irrigation.

It is well-known that biomass production represents a crucial factor in all agricultural systems; however, for turfgrass species, it is not as important as for other open field crops in terms of crop yield [1]. In fact, for landscape plants, such as bermudagrass, the visual appearance is much more important than biomass [64]. In a previous study, it was found that the visual quality of turf was not directly related to biomass production [65], although it is worth pointing out that the qualitative characteristics of a turf may deteriorate when plant growth is reduced [66]. Finally, from an economic point of view, the rate of herbeceaous biomass production in a given period of time should be related to the agronomic management of a turf. In fact, as affirmed by Zalacáin et al. [1], an increase in biomass production could generate a series of problems such us a greater number of mowings per year and the need to manage a higher volume of grass.

## Nutritional input by treated wastewater irrigation

In Table 9, the nutritional inputs of N, P and K provided by TWW irrigation with respect to conventional fertilisation programme are reported. It is well-know that TWW application provides macro and micronutrients for plant uptake and that these nutrients are present in forms which are easily available for plants, as well explained by various authors [13, 67]. In our study, we only took N, P and K into consideration in order to determine the degree to which they contributed to satisfy the nutrient demand of bermudagrass plants. TN, TP and K inputs provided by TWW irrigation were estimated by considering the average amount of these nutrients in TWW for each month of growth of bermudagrass plants. Other external sources of

**Table 9. Nitrogen, phosphorus and potassium inputs provided by treated wastewater irrigation during 2016–2018.** For each planted unit and month of growth, average values (± standard deviation) are shown.

| Parameter | Conventional fertilisation (kg ha$^{-1}$) | Nutrient input with TWW irrigation | |
|---|---|---|---|
| | | TWW[1] (kg ha$^{-1}$) | TWW[2] (kg ha$^{-1}$) |
| Total Nitrogen (TN) | | | |
| May | 50.0 | 10.4 ± 1.6 | 10.6 ± 1.44 |
| June | 50.0 | 9.93 ± 1.09 | 10.1 ± 1.09 |
| July | 50.0 | 9.15 ± 1.13 | 9.51 ± 1.15 |
| August | 50.0 | 10.0 ± 2.30 | 9.63 ± 1.27 |
| September | 50.0 | 10.5 ± 1.70 | 9.84 ± 1.06 |
| Total Phosphorus (TP) | | | |
| May | 20.0 | 4.72 ± 0.41 | 5.03 ± 0.80 |
| June | 20.0 | 4.86 ± 0.43 | 5.16 ± 0.76 |
| July | 200 | 4.73 ± 0.15 | 4.88 ± 0.87 |
| August | 10.0 | 4.52 ± 0.25 | 5.02 ± 1.02 |
| September | 10.0 | 4.83 ± 0.35 | 5.23 ± 0.83 |
| Potassium (K) | | | |
| May | 40.0 | 27.3 ± 1.82 | 35.1 ± 5.29 |
| June | 40.0 | 25.5 ± 2.10 | 28.5 ± 3.06 |
| July | 40.0 | 24.1 ± 3.63 | 36.1 ± 8.06 |
| August | 40.0 | 26.8 ± 2.17 | 28.8 ± 2.58 |
| September | 40.0 | 28.6 ± 1.46 | 30.3 ± 3.78 |

[1] TWW from giant reed-unit.

[2] TWW from umbrella sedge-unit.

nutrients, such as those obtained from organic matter decomposition were not assessed. Our results highlight that TWW irrigation allowed a saving of approximately 50.0 kg TN ha$^{-1}$ year$^{-1}$, 24.0 kg TP ha$^{-1}$ year$^{-1}$ and 29.0 kg P ha$^{-1}$ year$^{-1}$ on average with respect to widely-used N (300.0 kg ha$^{-1}$ year$^{-1}$), P (100.0 kg ha$^{-1}$ year$^{-1}$) and K (200.0 kg ha$^{-1}$ year$^{-1}$) fertilisation programmes for this species in the Mediterranean region, as documented by literature [36, 37].

This means that the use of TWW provides essential nutrients for plant growth and a considerable reduction in the use of mineral fertiliser, in general. This aspect was previously studied by a number of authors [1, 68, 69] who highlight a series of benefits, such as a reduction in fertiliser costs and the prevention of soil contamination in the long-term. On the contrary, some authors [70, 71] sustain that the use of TWW can determine a significant accumulation of soluble macro and micronutrients in the soil over time and negatively affects plant growth and soil quality. However, as sustained by Fan et. [68], this effect depends on a number of factors such as climate conditions, soil type and turf species. In particular, for several warm-season turfgrasses, no negative effects on morphological traits were found [72] following prolonged saline irrigation and salt accumulation. It was also documented [27] that most soil and plant related problems regarding the use of WW can have a lower environmental and economic effect on turfgrass growth than on food crops. Considering the benefits provided by TWW in terms of water and nutrient supply, and given the high water and fertiliser needs of bermudagrass in the Mediterranean area, on the basis of our results, it possible to affirm that TWW irrigation contributes to the sustainable management of turf without causing negative effects on plants in the medium term.

## Soil chemical characteristics

Soil chemical characteristics (S6 Table) as affected by year and irrigation water are shown in Table 10. The main factors did not determine significant variations for all parameters in the study. The year-by-irrigation water interaction was significant for TOC, TKN, P and Na. Soil pH varied very little during the 3-year period. However, no significant changes in topsoil pH of the FW- and TWW-irrigated soil were found. These results were in accordance with findings in literature [17, 18, 26, 73–75] which reported the inconsistent effect of TWW irrigation on soil pH in short- and medium-term. In particular, some authors [73] took the buffering action of the soil into consideration to explain the low variation of soil pH, highlighting that application of TWW could significantly affect it in the long-term. Soil salinity increased significantly over the years and was detected to be higher in TWW-irrigated soil (+6.3%) in comparison with FW-irrigated soil. Higher EC values were recorded subsequently in TWW-irrigated soil due to higher amounts of total dissolved salts in TWW and contributed to increasing salt levels in the soil. The higher accumulation of salts in TWW-irrigated soil can be explained by considering also the high percentage of clay in the soil structure and its relative cation-exchange capacity.

In the same way, TOC increased over the years and was found to be higher in TWW-irrigated soil (+2.90%) than FW-irrigated soil. As reported in a previous study [74], it is reasonable to claim that TWW irrigation significantly affects the topsoil organic matter based on irrigation duration and the amount of organic compounds in the TWW. Our results were fully confirmed by a number of authors [13, 17, 24, 26, 75] who investigated how TWW influenced the soil characteristics at different times.

Concerning mineral nutrients, such as N, P, K, Ca and Mg, it is evident that their accumulation in the topsoil can be linked to the original levels in FW and TWW, but these soil levels tend to change due to irrigation duration and the main actions conducted by plants and microorganisms [75]. In the case of N, the fact that significant differences were found between FW-and TWW-irrigated soils could be due to inconsistent effects of leaching and plant uptake on N content in the medium-term.

In our study, Na content increased over the 3 years and TWW-irrigated soil had higher Na content compared to FW-irrigated soil. It is well-known that an excess of Na in the soil can displace divalent cations, such as Ca and Mg, leading to soil structure deterioration [26]. As a consequence, periodic applications of good quality irrigation water seems necessary to avoid any risk to soil structure. Despite higher Na levels in TWW-irrigated soil, it is worth noting

**Table 10. Chemical characteristics of soil in response to year and irrigation water during the study period.** Average values of 3-year tests are shown.

|  | pH | EC ($\mu$S cm$^{-1}$) | TOC (g kg$^{-1}$) | TKN (g kg$^{-1}$) | P (mg kg$^{-1}$) | Total CaCO$_3$ (g kg$^{-1}$) | K (ppm) | Mg (ppm) | Na (ppm) |
|---|---|---|---|---|---|---|---|---|---|
| Year (Y) |  |  |  |  |  |  |  |  |  |
| Y$_1$ | 7.66 A | 191.3 B | 7.78 B | 1.27 C | 31.7 B | 1.33 B | 555.9 B | 639.1 A | 93.7 B |
| Y$_2$ | 7.63 B | 200.5 A | 7.82 B | 1.29 B | 31.5 B | 1.33 B | 560.9 AB | 637.5 A | 92.9 B |
| Y$_3$ | 7.65 A | 202.4 A | 8.0 A | 1.35 A | 32.4 A | 1.38 A | 563.9 A | 644.1 A | 97.2 A |
| Irrigation Water (IW) |  |  |  |  |  |  |  |  |  |
| IW$_1$ | 7.65 A | 190.8 B | 7.72 B | 1.26 C | 31.5 B | 1.34 B | 543.4 B | 634.6 B | 90.8 B |
| IW$_2$ | 7.65 A | 200.3 A | 7.91 A | 1.31 B | 32.5 A | 1.36 A | 568.2 A | 641.4 AB | 96.7 A |
| IW$_3$ | 7.65 A | 203.2 A | 7.94 A | 1.34 A | 31.6 B | 1.35 AB | 569.2 A | 644.8 A | 96.3 A |
| Y×IW | n.s. | n.s. | ** | ** | ** | n.s. | n.s. | n.s. | ** |

Means followed by the same letter in the same column are not significantly different according to Tukey's test (p $\leq$ 0.01).

** significant at p $\leq$ 0.01; n.s. = not significant.

that average SAR values (Table 5) were found below the values that could negatively affect soil properties (SAR > 10).

## Conclusions

In regions with prolonged water scarcity, TWW irrigation represents a sustainable practice to satisfy the water and nutrient needs of turfgrass species. Bermudagrass is the most important warm-season species in the Mediterranean area and usually requires high water and nutritional inputs to grow. This study demonstrates that the application of TWW irrigation in the medium-term (a time interval between two and ten years) leads to increases in bermudagrass turf biomass yields but does not affect the aesthetic value of the plants, when compared to conventional irrigation. This is important in terms of maintaining good quality turf but could determine a series of problems such as, increases in mowing requirements, greater quantities of biomass to be disposed of or reused and greater labor requirements. On the other hand, as highlighted by results, TWW irrigation leads to save in nutrients with respect to commonly-used fertilisation programmes and to benefits in terms of mineral fertilisers costs and environmental pollution. TWW can be a dangerous source of Na in the medium term, especially when irrigation is prolonged over the time. Therefore, to avoid any inconvenience for plant and soil, any adequate agronomic practices are encouraged. We conclude that TWW represents a useful element of sustainable agriculture in the management of open field crops; however, despite the benefits, application must be monitored over the time to prevent any possible damage.

## Supporting information

**S1 Fig. An overview of the pilot-scale HSSFs CW.**
(PDF)

**S2 Fig. Layout of the wastewater treatment system based on HSSFs CW.**
(PDF)

**S1 Table. Main dataset of climate data in the experimental site.**
(XLSX)

**S2 Table. Main average dataset of the TWW chemical and microbiological parameters of the giant reed-planted unit (HSSF CW) for the analyses.**
(XLS)

**S3 Table. Main average dataset of the TWW chemical and microbiological parameters of the umbrella sedge-planted unit (HSSF CW) for the analyses.**
(XLS)

**S4 Table. Main 10-day average dataset of $ET_c$, $Q_i$ and $Q_0$ in the pilot HSSFs CW for the analyses.**
(XLSX)

**S5 Table. Main 3-year average dataset related to morphological, qualitative and production parameters of bermudagrass plants irrigated with FW and TWW.**
(XLS)

**S6 Table. Main 3-year average dataset related to pH, organic compounds, nutrients and salts content in FW and TWW-irrigated soils.**
(XLS)

## Acknowledgments

The authors would like to thank the Sicilian Regional Ministry of Food and Agricultural Resources, the CoRiSSIA Research Centre of Palermo and the University of Palermo. A special thanks goes to Branwen Hornsby for her continuing support to the research and linguistic contribution. The authors would also like to thank the reviewers for their constructive comments.

## Author Contributions

**Conceptualization:** Nicolò Iacuzzi.

**Data curation:** Davide Farruggia.

**Formal analysis:** Davide Farruggia.

**Funding acquisition:** Claudio Leto.

**Investigation:** Mario Licata, Giuseppe Di Miceli.

**Methodology:** Mario Licata.

**Project administration:** Claudio Leto.

**Resources:** Claudio Leto, Teresa Tuttolomondo.

**Software:** Davide Farruggia.

**Supervision:** Teresa Tuttolomondo.

**Validation:** Nicolò Iacuzzi, Teresa Tuttolomondo, Giuseppe Di Miceli.

**Visualization:** Teresa Tuttolomondo, Giuseppe Di Miceli.

**Writing – original draft:** Mario Licata, Davide Farruggia.

**Writing – review & editing:** Mario Licata, Nicolò Iacuzzi, Giuseppe Di Miceli.

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
