## [Decision Letter · Decision Letter 0]

11 May 2022

PONE-D-22-09242Irrigation with treated wastewater affects biomass production of bermudagrass and reduces the need for plant nutrientsPLOS ONE

Dear Dr. Iacuzzi,

Thank you for submitting your manuscript to PLOS ONE. After careful consideration, we feel that it has merit but does not fully meet PLOS ONE’s publication criteria as it currently stands. Therefore, we invite you to submit a revised version of the manuscript that addresses the points raised during the review process.

We look forward to receiving your revised manuscript.

Kind regards,

Vassilis G. Aschonitis

Academic Editor

PLOS ONE

Journal Requirements:

[The authors would like to thank the Sicilian Regional Ministry of Food and Agricultural Resources, the Corissia Research Centre and the University of Palermo for having provided the funds for the study. A special thanks also goes to Branwen Hornsby for her continuing support to the research and linguistic contribution.]

 [The funders had no role in study design, data collection and analysis, decision to publish, or preparation of the manuscript.]

Reviewers' comments:

Reviewer's Responses to Questions

**Comments to the Author**

1. Is the manuscript technically sound, and do the data support the conclusions?

Reviewer #1: Partly

Reviewer #2: Yes

2. Has the statistical analysis been performed appropriately and rigorously? 

Reviewer #1: No

Reviewer #2: Yes

3. Have the authors made all data underlying the findings in their manuscript fully available?

Reviewer #1: No

Reviewer #2: Yes

4. Is the manuscript presented in an intelligible fashion and written in standard English?

Reviewer #1: No

Reviewer #2: Yes

5. Review Comments to the Author

Reviewer #1: Reviewer comments

The manuscript entitled ‘Irrigation with treated wastewater affects biomass production of bermudagrass and reduces the need for plant nutrients’ is an interesting research work. The article is not well organized with robust bibliography which consists of enough data and enough scientific fundamental conceptualization. The article is not suitable for publication in PLOSONE.

• Title of the manuscript need to be improved. In the present form it appears as the statement. Authors may consider inclusion of scientific name of bermudagrass in brackets ().

• Authors are advised to reduce the data values (large values) to single decimal place throughout the manuscript. In abstract, please check, …….29.00 kg P ha-1 year-1, I think should be K rather than P????

• Authors have written several abbreviations e.g. BOD, COD, TN, TP etc, which needs to written in full at their first appearance in the text.

• In abstract authors have written that soil salinity increased significantly…., they need to highlight the statistical significance e.g. p<0.05 or p<0.01.

• The introduction has citation of 32 references, of which only 5-6 references are latest one (work published after 2019). Authors are advised to cite recent references.

• The major drawback of the introduction section is the lack of state-of the art of the research works and its connection with the problem formulation, research objectives and the hypothesis. Authors should critically work on that.

• Authors need to specify the amount of wastewater for irrigation per hectare area or number of irrigations to be given with wastewater per annum. It is written in conclusion that application of TWW irrigation in medium-term leads to increase in bermudagrass turf biomass yields. Clarify in conclusion what does medium term means. Conclusion must be clear.

• Figure 1 and 2 are needless.

• The quality of figures 4 and 5 is very bad. It is difficult to read the legends.

• Figure 6 lacks statistical analysis

Reviewer #2: Respected authors

Following are the comments please consider them

1) Topic- Make title in more questionable way. Your title looks conclusive as it concludes the results in it.

Suggestion- Effect of treated wastewater irrigation on soil properties and bermuda grass production.

2) otherwise manuscript is well constructed and written in beautiful manner

6. PLOS authors have the option to publish the peer review history of their article (what does this mean?). If published, this will include your full peer review and any attached files.

Reviewer #1: No

Reviewer #2: **Yes: **Harmandeep Singh Chahal

---

## [Author Response · Author response to Decision Letter 0]

31 May 2022

Revision notes

Ms. Ref. No.: PONE-S-22-11993

In response to Ms. Ref. No.: PONE-S-22-11993 we followed all the recommendations which were made by the editor and reviewers. 

Editor’s comments

Point 1: Please ensure that your manuscript meets PLOS ONE's style requirements, including those for file naming. The PLOS ONE style templates can be found at https://journals.plos.org/plosone/s/file?id=wjVg/PLOSOne_formatting_sample_main_body.pdf and 

Response: the authors confirm that the manuscripts meets PLOS ONE’s style requirement. They have used the PLOS ONE templates, found at links suggested by the Editor.

Point 2: We note that the grant information you provided in the ‘Funding Information’ and ‘Financial Disclosure’ sections do not match. When you resubmit, please ensure that you provide the correct grant numbers for the awards you received for your study in the ‘Funding Information’ section.

Response: the authors have provided the correct grant number for the awards we received in our study in the “Funding Information” section, as suggested by the Editor. At this stage, they cannot include the grant information also in the “Financial Disclosure” due to fact that they cannot view it in the submission platform of Plos One. If you want, you could kindly insert this information in the Financial Disclosure as the following:

Title of the project: Tecnologie innovative per l'impiego di acque "non convenzionali" e prevenzione della desertificazione.

Grant numbers D71D04000000008

Full name of the funder: Ministry of Agriculture, Rural Development and Mediterranean Fisheries of Sicilian Region

Point 3: Thank you for stating the following in the Acknowledgments Section of your manuscript: [The authors would like to thank the Sicilian Regional Ministry of Food and Agricultural Resources, the Corissia Research Centre and the University of Palermo for having provided the funds for the study. A special thanks also goes to Branwen Hornsby for her continuing support to the research and linguistic contribution.] We note that you have provided funding information that is not currently declared in your Funding Statement. However, funding information should not appear in the Acknowledgments section or other areas of your manuscript. We will only publish funding information present in the Funding Statement section of the online submission form. Please remove any funding-related text from the manuscript and let us know how you would like to update your Funding Statement. Currently, your Funding Statement reads as follows: 

 [The funders had no role in study design, data collection and analysis, decision to publish, or preparation of the manuscript.] Please include your amended statements within your cover letter; we will change the online submission form on your behalf.

Response: the authors have removed any funding-related text from the manuscript, as suggested by the Editor. Furthermore, the authors have included the amended statements within the cover letter.

Point 4: We note that you have stated that you will provide repository information for your data at acceptance. Should your manuscript be accepted for publication, we will hold it until you provide the relevant accession numbers or DOIs necessary to access your data. If you wish to make changes to your Data Availability statement, please describe these changes in your cover letter and we will update your Data Availability statement to reflect the information you provide.

Response: the authors agree with the Editor’s comment. They have uploaded a minimal data sets which underline the results more relevant in the manuscript as Supporting Information files, as requested by the Journal (all relevant data are within the manuscript and its Supporting Information files).

Point 5: We note that you have included the phrase “data not shown” in your manuscript. Unfortunately, this does not meet our data sharing requirements. PLOS does not permit references to inaccessible data. We require that authors provide all relevant data within the paper, Supporting Information files, or in an acceptable, public repository. Please add a citation to support this phrase or upload the data that corresponds with these findings to a stable repository (such as Figshare or Dryad) and provide and URLs, DOIs, or accession numbers that may be used to access these data. Or, if the data are not a core part of the research being presented in your study, we ask that you remove the phrase that refers to these data.

Response: the authors agree with the Editor’s comment. They have decided to remove the phrase that refers to these data.

Reviewer #1

In response to Ms. Ref. No.: PONE-S-22-11993 Reviewer #1, we tried to follow nearly all of his/her recommendations. Here is a point by point summary of the actions taken in response to the reviewer’s comments.

Point 1: The manuscript entitled ‘Irrigation with treated wastewater affects biomass production of bermudagrass and reduces the need for plant nutrients’ is an interesting research work. The article is not well organized with robust bibliography which consists of enough data and enough scientific fundamental conceptualization. The article is not suitable for publication in PLOSONE.

Response: the authors appreciate and thank the reviewer for his/her comments. They have improved the manuscript following the suggestions of the reviewer.

Point 2: Title of the manuscript need to be improved. In the present form it appears as the statement. Authors may consider inclusion of scientific name of bermudagrass in brackets (). 

Response: the authors agree with the reviewer’s consideration They have changed the title of the manuscript and included the scientific name of bermudagrass in brackets, as suggested by the reviewer.

Point 3: Authors are advised to reduce the data values (large values) to single decimal place throughout the manuscript. In abstract, please check, …….29.00 kg P ha-1 year-1, I think should be K rather than P????

Response: the authors have reduced the large data values (values above 10) to single decimal place as suggested by the reviewer. They apologize with him/her for having written P instead of K in the abstract; then, they have made a small correction.

Point 4: Authors have written several abbreviations e.g. BOD, COD, TN, TP etc, which needs to written in full at their first appearance in the text.

Response: the authors agree with the reviewer’s observation. They have written BOD, COD, TN, TP etc, in full at their first appearance in the text, as suggested by the reviewer.

Point 5: In abstract authors have written that soil salinity increased significantly…., they need to highlight the statistical significance e.g. p<0.05 or p<0.01.

Response: the authors have highlighted the statistical significance (p<0.01) in the abstract, as suggested by the reviewer.

Point 6: The introduction has citation of 32 references, of which only 5-6 references are latest one (work published after 2019). Authors are advised to cite recent references.

Response: the authors agree with the reviewer’s observation. They have added new references and cited recent references (where possible), as suggested by the reviewer. 

Point 7: The major drawback of the introduction section is the lack of state-of the art of the research works and its connection with the problem formulation, research objectives and the hypothesis. Authors should critically work on that.

Response: the authors thank the reviewer for his/her constructive comment. They have improved the introduction, added references in order to highlight the state of the art of the research topic and included the hypothesis. 

Point 8: Authors need to specify the amount of wastewater for irrigation per hectare area or number of irrigations to be given with wastewater per annum. It is written in conclusion that application of TWW irrigation in medium-term leads to increase in bermudagrass turf biomass yields. Clarify in conclusion what does medium term means. Conclusion must be clear.

Response: the authors agree with the reviewer’s comment. They have specified in the manuscript the amount of treated wastewater for irrigation per hectare area, as suggested by the reviewer. Concerning the number of irrigations to be given with treated wastewater per year, they had already provided this information in the first version of the manuscript (…in 2016-2018, irrigation was applied from May to September three times per week, on average, both with FW and TWW..).

Furthermore, they have clarified in the conclusions what medium term means: it is a time interval between two and ten years, as many researchers intend.

Point 9: Figure 1 and 2 are needless.

Response: the authors have deleted the figures 1 and 2 from the manuscript, as suggested by the reviewer.

Point 10: the quality of figures 4 and 5 is very bad. It is difficult to read the legends.

Response: the authors have improved the quality of figures 4 and 5 and increased the character of the legends in order to make them more readable, as suggested by the reviewer. The image resolution of the two figures is now 600 dpi. I kindly advise the reviewer that the quality of the image could deteriorate due to the transformation from TIF to PDF. I hope the reviewer understand this observation.

Point 11: Figure 6 lacks statistical analysis

Response: the authors highlight that, in the case of figure 6, they did not carry out the analysis of the variance but they analysed the variability of data using standard deviation calculation. In Figure 6, the bars of standard deviation are, in fact, showed. The authors have deleted “significantly” from the text in the paragraph (to avoid any confusion) concerning Figure 6.

Reviewer #2

In response to Ms. Ref. No.: PONE-S-22-11993 Reviewer #2, we tried to follow nearly all of his/her recommendations. Here is a point by point summary of the actions taken in response to the reviewer’s comments.

Point 1: Topic- Make title in more questionable way. Your title looks conclusive as it concludes the results in it. Suggestion- Effect of treated wastewater irrigation on soil properties and bermudagrass production.

Response: the authors agree with the reviewer’s consideration. They have changed the title of the manuscript, as suggested by the reviewer.

Point 2: otherwise manuscript is well constructed and written in beautiful manner

Response: the authors appreciate and thank the reviewer for his/her comments.

We hope the editorial board will agree on the interest of this study.

Sincerely yours,

Nicolò Iacuzzi on behalf of the authors.

Corresponding author: Dr Nicolò Iacuzzi, Department of Agricultural, Food and Forest Sciences, Università degli Studi di Palermo, Viale delle Scienze 13 Building 4, 90128 Palermo, Italy. E-mail: nicolo.iacuzzi@unipa.it

---

## [Decision Letter · Decision Letter 1]

5 Jul 2022

Effect of irrigation with treated wastewater on bermudagrass (Cynodon dactylon (L.) Pers.) production and soil characteristics and estimation of plant nutritional input

PONE-D-22-09242R1

Dear Dr. Iacuzzi,

We’re pleased to inform you that your manuscript has been judged scientifically suitable for publication and will be formally accepted for publication once it meets all outstanding technical requirements.

Kind regards,

Vassilis G. Aschonitis

Academic Editor

PLOS ONE

Additional Editor Comments (optional):

Reviewers' comments:

Reviewer's Responses to Questions

**Comments to the Author**

1. If the authors have adequately addressed your comments raised in a previous round of review and you feel that this manuscript is now acceptable for publication, you may indicate that here to bypass the “Comments to the Author” section, enter your conflict of interest statement in the “Confidential to Editor” section, and submit your "Accept" recommendation.

Reviewer #3: All comments have been addressed

2. Is the manuscript technically sound, and do the data support the conclusions?

Reviewer #3: Yes

3. Has the statistical analysis been performed appropriately and rigorously? 

Reviewer #3: Yes

4. Have the authors made all data underlying the findings in their manuscript fully available?

Reviewer #3: Yes

5. Is the manuscript presented in an intelligible fashion and written in standard English?

Reviewer #3: Yes

6. Review Comments to the Author

Reviewer #3: All the corrections have been done. All the comments have been answered. The manuscript can be pubblished. No further comments

7. PLOS authors have the option to publish the peer review history of their article (what does this mean?). If published, this will include your full peer review and any attached files.

Reviewer #3: No

---

## [Editor Report · Acceptance letter]

7 Jul 2022

PONE-D-22-09242R1 

Effect of irrigation with treated wastewater on bermudagrass (*Cynodon dactylon* (L.) Pers.) production and soil characteristics and estimation of plant nutritional input 

Dear Dr. Iacuzzi:

I'm pleased to inform you that your manuscript has been deemed suitable for publication in PLOS ONE. Congratulations! Your manuscript is now with our production department. 

Kind regards, 

on behalf of

Dr. Vassilis G. Aschonitis 

Academic Editor

PLOS ONE